# Theoretical Importance of PVP-Alginate Hydrogels Structure on Drug Release Kinetics

**DOI:** 10.3390/gels5020022

**Published:** 2019-04-18

**Authors:** Michela Abrami, Paolo Marizza, Francesca Zecchin, Paolo Bertoncin, Domenico Marson, Romano Lapasin, Filomena de Riso, Paola Posocco, Gabriele Grassi, Mario Grassi

**Affiliations:** 1Department of Engineering and Architecture, Trieste University, via Valerio 6, I-34127 Trieste, Italy; michela.abrami@dia.units.it (M.A.); paomariz83@gmail.com (P.M.); fra_zecchin@hotmail.it (F.Z.); domenico.marson@dia.units.it (D.M.); romano.lapasin@dia.units.it (R.L.); paola.posocco@dia.units.it (P.P.); 2Department of Life Sciences, University of Trieste, Piazzale Europa 1, I-34127 Trieste, Italy; pbertoncin@units.it; 3Department of Micro- and Nanotechnology, Technical University of Denmark (DTU), Ørsteds Plads Bygning 345Ø, 2800 Kgs Lyngby, Denmark; filomenaderiso@hotmail.it; 4Department of Life Sciences, Cattinara University Hospital, Trieste University, Strada di Fiume 447, I-34149 Trieste, Italy; ggrassi@units.it

**Keywords:** mathematical modelling, hydrogel, drug release, rheology, low field NMR

## Abstract

Background: The new concepts of personalized and precision medicine require the design of more and more refined delivery systems. In this frame, hydrogels can play a very important role as they represent the best surrogate of soft living tissues for what concerns rheological properties. Thus, this paper focusses on a global theoretical approach able to describe how hydrogel polymeric networks can affect the release kinetics of drugs characterized by different sizes. The attention is focused on a case study dealing with an interpenetrated hydrogel made up by alginate and poly(*N*-vinyl-2-pyrrolidone). Methods: Information about polymeric network characteristics (mesh size distribution and polymer volume fraction) is deduced from the theoretical interpretation of the rheological and the low field Nuclear Magnetic Resonance (NMR) characterization of hydrogels. This information is then, embodied in the mass balance equation whose resolution provides the release kinetics. Results: Our simulations indicate the influence of network characteristics on release kinetics. In addition, the reliability of the proposed approach is supported by the comparison of the model outcome with experimental release data. Conclusions: This study underlines the necessity of a global theoretical approach in order to design reliable delivery systems based on hydrogels.

## 1. Introduction

Hydrogels are coherent systems composed of a three-dimensional network, usually of polymeric origin, containing a huge amount of a continuous aqueous phase (sometimes exceeding 99% v/v), which cannot dissolve the network due to the presence of inter-fiber connections, called crosslinks [1]. Interestingly, despite the huge water volume fraction, hydrogels show rheological behaviors closer to that of solids rather than to that of liquids [1] and this is the reason they represent the best surrogate of soft living tissues for what concerns the rheological behavior [2]. Usually, hydrogels classification relies on the crosslink nature, fiber origin, composition, and charge [3]. From a crosslinking point of view, hydrogels are defined as chemical when crosslinks between different chains (fibers) are strong, permanent, and punctual, due to the presence of covalent bonds. Conversely, they are defined as physical when crosslinks are represented by either chains of topological entanglements (in this case, a pseudo-gel takes place as entanglements have a short lifetime so that a temporary network occurs) or physical interactions, such as H-bonds, ionic, Coulombic, van der Waals, dipole–dipole, and hydrophobic interactions. Regarding their origins, hydrogels can be natural or synthetic depending on the nature of the fibers constituting the network. Typical natural hydrogels are based on agar, collagen, chitosan, alginate, hyaluronic acid, gelatin, and fibrin [4,5]. On the contrary, some examples of synthetic hydrogels are those made up by d,l-lactide-*co*-glycolide (PLGA), polyamidoamine (PAMAM), poly(caprolactone-*co*-ethylethylene phosphate) (PCLEEP), and poly(*N*-vinyl-2-pyrrolidone) (PVP). An interesting hydrogels class is represented by an interpenetrating polymeric network (IPN) that is made up of two (or more) independent cross-linked synthetic and/or natural polymeric chains [6]. Finally, hydrogels can be nonionic, ionic, ampholytic, or zwitterionic, depending on the polarization features of the polymeric chains constituting the network.

Drug release from hydrogels is usually controlled by physical, physicochemical, and system related factors [7]. While swelling/shrinking and surface erosion represent physical factors, physicochemical factors comprehend bulk erosion, drug dissolution (recrystallization), drug transport (by diffusion and convection), and drug interaction with the hydrogel network. Finally, initial distribution of the drug concentration inside the hydrogel, hydrogel geometry (cylindrical, spherical, etc.), and size distribution (this is the case of polydispersed ensembles of hydrogels) represent system related mechanisms. Typically, the variations of external factors, such as temperature, pH, and fluid dynamics, are responsible for the swelling/shrinking processes. Indeed, these changes force the system to attain new equilibrium conditions as they induce a variation of the water chemical potential inside and outside the hydrogel. In particular, this occurs when a dry hydrogel is put in contact with an external aqueous environment [8]. Erosion may occur due to chemical (hydrolytic/enzymatic degradation of polymeric chains) and/or physical (chains disentanglement due to the hydrodynamic conditions of the external aqueous environment) factors. It is superficial when only the hydrogel surface is affected and massive or “bulk” when it involves the whole hydrogel volume [9]. For stability reasons, hydrogels are stored dry, and drug release begins only if an external aqueous fluid swells the polymeric network, allowing drug dissolution and diffusion through the enlarged meshes of the polymeric network. In that case, drug dissolution represents a fundamental step. Indeed, when metastable compounds, like polymorphs, amorphous, or nano-crystalline drugs, are present in the dry hydrogel, the dissolution process can be affected by recrystallization, which leads to the formation of a new, more stable drug crystalline organization induced by the contact with the entering water [10]. As drug solubility typically decreases upon the formation of the more stable drug form, recrystallization can play a very important role in the kinetics of the whole release process [11,12] and on drug bioavailability [13]. Last, but not least, network mesh size distribution and possible drug interaction with the three-dimensional network can strongly affect the kinetics of drug release [14]. For example, electrostatic interactions, typically occurring between charged polypeptides/antibiotics and collagen, can be the reason of drug adsorption/desorption on the three-dimensional network [15]. In addition, drug–network interactions may be due to hydrogen bonds [16], lipophilic [17] as well as non-covalent interactions as occurs in imprinted polymeric networks [18].

Although many factors affect drug release from hydrogels, in this paper, the attention is focused on the hindering effect exerted by the polymeric network on drug diffusion. Thus, we implicitly assume that all other phenomena play a marginal role, this being the common case of a hydrogel that (1) does not undergo swelling/erosion, (2) that is loaded by a drug that does not interact with the network fibers, and (3) that can host the drug in its solubilized form inside the fluid phase distributed in the entire hydrogel network. In this frame, it is clear that the determination of the mesh size distribution of the three-dimensional network plays a predominant role. Among the different ways by which mesh size distribution can be determined [19], we propose the synergistic combination of rheology and low field Nuclear Magnetic Resonance (LF-NMR). Consequently, the aim of this paper is to adopt an overall theoretical approach based (1) on the determination of the polymeric network mesh size distribution and (2) on the evaluation of its effect on the diffusion driven drug release process. As a case study, an IPN made up by poly(*N*-vinyl-2-pyrrolidone) (PVP) and alginate at different polymer mass percentages is considered [20]. The IPN was prepared by the sequential reticulations of PVP and alginate induced by the exposure to ultraviolet (UV) light and to calcium ions.

## 2. Results and Discussion

Stress sweep tests revealed (data not shown) that, for all the studied hydrogels, the linear viscoelastic range holds for stresses above the value adopted for the frequency sweep tests (τ = 5 Pa). In addition, Figure 1a reports the results of the frequency sweep test (mechanical spectra) referring to systems characterized by a polymer mass percentage equal to 20.5% (20% PVP and 0.5% alginate) and three different irradiation times (A—22 min; B—33 min; C—44 min). Similarly, Figure 1b shows the mechanical spectra referring to systems characterized by a polymer mass percentage equal to 30.5% (30% PVP and 0.5% alginate) and three different irradiation times (A—22 min; B—33 min; C—44 min).

All the systems can be defined as gels, as the storage (*G*′) modulus is substantially independent of angular frequency, ω = 2π*f*, and sensibly higher (more than 10 times) than the loss modulus, *G*″ [1]. In addition, Figure 1a,b proves that the increase of the irradiation time results in higher *G*′ values for both 20 and 30 hydrogels even if hydrogel 20A and 20B show very similar *G*′ values. Increasing polymer concentrations, for an equal irradiation time, likewise results in higher *G*′ values. The mechanical spectra were fitted to the generalized Maxwell model consisting of a parallel combination of Maxwell elements and a pure elastic element (g_e_). Accordingly, the dependence of the elastic (*G*′) and viscous (*G*″) moduli on pulsation, ω = 2π*f*, is given by the following expressions:
(1)$$G^{\prime }=g_{e}+\sum_{i=1}^{n_{R}}g_{i}\frac{{\left(\lambda_{i}\omega \right)}^{2}}{1+{\left(\lambda_{i}\omega \right)}^{2}}$$
(2)$$G^{\prime \prime }=\sum_{i=1}^{n_{R}}g_{i}\frac{\left(\lambda_{i}\omega \right)}{1+{\left(\lambda_{i}\omega \right)}^{2}}$$where $g_{i}=\eta_{i}/\lambda_{i}$, *n*_R_ is the number of Maxwell elements considered, *g*_i_, η_i_, and λ_i_ represent, respectively, the elastic modulus, the viscosity, and the relaxation time of the ith Maxwell element. The equilibrium modulus, *g*_e_, measures the contribution of the purely elastic element. The simultaneous fitting of Equations (1) and (2) to experimental *G*′ and *G*″ data was performed assuming that relaxation times (λ_i_) were scaled by a factor 10 (λ_i+1_ = 10λ_i_) [1]. Hence, the parameters of the model were 2 + *n*_R_ (i.e., λ_1_, *g*_e_ + *g*_i_ (1 ≤ *i* ≤ *n*_R_)). Based on a statistical procedure [21], *n*_R_ was selected in order to minimize the product, χ^2^*(2 + *n*_R_), where χ^2^ is the sum of the squared errors. Three to four Maxwell elements plus the purely elastic element guarantee a statistically reliable fitting (positive F-test). Table 1 reports the fitting parameters values. 

Assuming that the shear modulus (*G*) is obtained by the sum of all *g*_i_ plus *g*_e_ [22], Flory theory [23] allows the determination of the polymeric network crosslink density (ρ_x_), defined as the moles of junction points between different polymeric chains per hydrogel unit volume:
(3)$$\rho_{x}=G/RT$$where *R* is the universal gas constant and *T* is the absolute temperature. The link between ρ_x_ and the average mesh size of the polymeric network (ξ) is provided by the equivalent network theory [24]. As it is very difficult to account for all the irregularities of a real polymeric network, this theory replaces the real network structure by an idealized one constituted by a perfect three-dimensional cubic arrangement of crosslinks. The common aspect of the real architecture and idealized one relies on the same average, ρ_x_. By labeling with ξ the space occurring between two consecutive crosslinks and using the crosslink density definition, it is possible to establish a simple connection between ξ and ρ_x_. Indeed, the volume ((4/3)π(ξ/2)^3^) associated to each crosslink in both architectures has to be equal to 1/(*N*_A_ ρ_x_), with *N*_A_ the Avogadro number. Consequently, ρ_x_ and ξ are connected through the following relation:
(4)$$\xi =\sqrt[{3}]{6/\pi \rho_{x}N_{A}}$$

Table 2 reports the values of *G*, ρ_x_, and ξ referring to the six hydrogels considered.

Knowing the average mesh size, it is possible to determine the mesh size distribution recurring to the LF-NMR characterization. Figure 2a,b shows the magnetic relaxation of the hydrogens belonging to the water molecules entrapped inside the six studied hydrogels. Regardless of the polymer concentration, higher irradiation times (*A* = 22 min, *B* = 33 min, *C* = 44 min) lead to a faster reduction of the modulus (*I*_s_) of the *xy* component of the global magnetization vector. On the contrary, at fixed irradiation times, the effect of the concentration seems insignificant for hydrogels *B* and *C*, while, hydrogel 30A is characterized by a slower relaxation time with respect to hydrogel 20A. In order to determine the relaxation times’ distribution (*A*_i_, *T*_2i_), the free induction decay (FID) time decay (*I*_s_) (see Figure 2a,b), related to the extinction of the *x*–*y* component of the magnetization vector, was fitted according to its theoretical estimation, *I*(*t*) [25]:
(5)$$I\left(t\right)=\sum_{i=1}^{m}A_{i}\exp({-t/T}_{2i})\;A_{i\%}=100*A_{i}/\sum_{i=1}^{m}A_{i}$$where *t* is time, *A*_i_ is the pre-exponential factor (dimensionless), proportional to the number of protons relaxing with the relaxation time, *T*_2i_, and *A*_i%_ is the percentage value of *A*_i_. The average relaxation time of protons, *T*_2m_, and its inverse value, (1/*T*_2_)_m_, are defined as:
(6)$$T_{2m}=\sum_{i=1}^{m}A_{i}T_{2i}/\sum_{i=1}^{m}A_{i}\;{\left(\frac{1}{T_{2}}\right)}_{m}=\sum_{i=1}^{m}\frac{A_{i}}{T_{2i}}/\sum_{i=1}^{m}A_{i}$$where *m* is the number of the relaxation times (*T*_2i_) constituting the relaxation time distribution (*A*_i_, *T*_2i_). *m* was determined by minimizing the product, χ^2^*(2*m*), where χ^2^ is the sum of the squared errors and 2*m* represents the number of fitting parameters of Equation (5) [21]. An inspection of Table 3, showing the values of the fitting parameters from Equation (5) makes it clear that two relaxation times (*T*_21_ and *T*_22_) are always necessary to describe the hydrogen relaxation of the six specimen studied.

The irradiation time and polymer concentration are expected to affect both the hydrogen relaxation time (*T*_2m_) and the polymeric network mesh size (ξ). Indeed, it is well known [25,26,27] that the average value of the inverse relaxation time (1/*T*_2_)*_m_* (see Equation (6)) referring to hydrogens belonging to water molecules trapped in the polymeric network is related to the polymer volume fraction, φ, and ξ, by the following equation:
(7)$${\left(\frac{1}{T_{2}}\right)}_{m}=\frac{1}{T_{2H2O}}+2\frac{M}{\xi \sqrt{\frac{C_{0}}{C_{1}}\;\frac{1-0.58\varphi }{\varphi }}}$$where *T*_2H2O_ is the relaxation time of the free water hydrogens (≈ 3008 ms at 25 °C, 20 MHz [28]), i.e., the relaxation time of water hydrogens in the absence of polymer network. M is a parameter, called relaxivity, representing the ratio between the thickness and the relaxation time of the water layer close to polymeric chains (bound water layer) while *C*_0_ and *C*_1_ are two constants that, for a cubical network, are equal to 1 and 3π, respectively. In addition, ξ is related to φ and to the polymeric chains radius, *R*_f_, by the following equation:
(8)$${\xi =R}_{f}\sqrt{\frac{C_{1}}{C_{0}}\frac{1-0.58\varphi }{\varphi }}$$

While ξ is known from the rheological characterization and (1/*T*_2_)_m_ can be evaluated by fitting Equation (5) to the experimental relaxation data shown in Figure 2a,b (see Table 3), φ is unknown. Indeed, not only does hydrogel realization imply a washing step, aimed to remove un-crosslinked polymer, but alginate ionotropic crosslinking induces hydrogel syneresis. Accordingly, the φ real value can be very different from its nominal value, as evaluated from the % polymer mass percentage. Thus, φ is evaluated on the basis of Equation (7), assuming that the M value referring to our hydrogels, composed by PVP and alginate, is the same as the pure PVP (7.7 × 10^−3^ nm/ms) [22]. This assumption is reasonable since the amount of alginate is very small compared to the PVP one (1/40 or 1/60). Then, Equation (8) allows the evaluation of *R*_f_. The relations between *T*_2m_, ξ, and φ descending from this analysis can be seen by looking at Figure 3.

As the distribution of the *T*_2m_, ξ, and φ values is Gaussian (as demonstrated by the Kolmogorov-Smirnov test), the analysis of their inter-correlations was performed according to the Pearson correlation coefficient, *r*. Interestingly, Figure 3 shows a strong inverse linear correlation between *T*_2m_ and φ (*r* = −0.976, *t*_r_(4, 0.95) < 8.5; correlation coefficient ρ = 0.92, *t*_ρ_(4, 0.95) < 4.6, F(4, 3, 0.95) > 2). On the contrary, no correlation exists between ξ and φ and between *T*_2m_ and ξ. The inverse linear correlation between *T*_2m_ and φ is theoretically correct as the φ increase implies the increase of the solid surface, represented by the polymeric chains’ surface, in contact with the water molecules. In turn, the solid surface is responsible for faster hydrogen relaxation [27]. On the contrary, the absence of a correlation between ξ and φ and between *T*_2m_ and ξ simply states that, whatever φ, the mesh size is almost constant. This is the result of the counteracting phenomena connected to polymer removal (due to hydrogel washing, see Section 4.3) and hydrogel syneresis caused by alginate ionotropic crosslinking. As a matter of fact, these phenomena imply a reduction of φ with respect to the theoretical value for hydrogels 30A, 20A, and 30B and a φ increase for the others (20B, 30C, and 20C).

LF-NMR characterization also allows the conclusion that, for all the six hydrogels, the average mesh size, ξ, is due to the existence of two different populations. The first is characterized by the relaxation time, *T*_21_, and relative abundance, *A*_1%_, and the second by the relaxation time, *T*_22_, and relative abundance, *A*_2%_ (see Table 3). In other words, assuming that M does not depend on the mesh size [25], Equation (7) holds also for each single population:
(9)$${\left(\frac{1}{T_{2}}\right)}_{i}=\frac{1}{T_{2H2O}}+2\frac{M}{\xi_{i}\sqrt{\frac{C_{0}}{C_{1}}\;\frac{1-0.58\varphi }{\varphi }}}$$where ξ_i_ and (1/*T*_2_)_i_ represent the mesh size and the inverse of the relaxation time pertaining to the population or class ith, respectively. Thus, Equation (9) allows an evaluation of ξ_1_ and ξ_2_, whose relative abundances are, respectively, *A*_1%_ and *A*_2%_ as reported in Table 4.

Figure 4 allows a comparison of the picture of the hydrogel nanostructure suggested by the rheological and LF-NMR characterization with that derived from a TEM picture of hydrogel 20B (see Section 4.4 for sample processing before taking the TEM picture). Given the purely qualitative value that TEM has in this context, Figure 4 shows the existence of a PVP network (grey ribbons), characterized by wide meshes (~100 nm), embedding cross-linked alginate agglomerates (dark zones), and featuring very small sized meshes (<10 nm). The structure is interpreted by the rheological and LF-NMR approach, in light of the equivalent network theory [24], with the existence of meshes spanning between 30 (80%) and 15 nm (20%) (see Table 4). As this estimation lies in between the mesh size bounds suggested by Figure 4, we believe it is physically acceptable.

Once network characteristics are known, it is necessary to embed them in a mass transport equation in order to evaluate their effect on the drug release kinetics. Among many other possible choices [29], we believe that the best way is to assume that the drug diffusion coefficient, *D*, depends on ξ and φ according to the Lustig-Peppas model [30]:
(10)$$D=D_{0}\left(1-\frac{{2r}_{s}}{\xi }\right)\exp(-Y\frac{\varphi }{1-\varphi })$$where *D*_0_ is the drug diffusion coefficient in pure water, *r*_s_ is the radius of the sphere embodying the drug molecule, and Y is a parameter that can be set to be equal to 1 for correlation purposes [30] even if, in general, it can take higher values [29,31]. Assuming Y = 1 in Equation (10), Figure 5 shows the variation of the ratio of *D*/*D*_0_ assuming drugs of different dimension (theophylline, *r*_s_ = 0.39 nm; vitamin B_12_, *r*_s_ = 0.86 nm; myoglobin, *r*_s_ = 1.89 nm; bovine serum albumin (BSA) *r*_s_ = 3.6 nm; immunoglobulin G *r*_s_ = 5.63 nm [32]) and the ξ–φ couples competing with our six hydrogels (see Figure 3 and Table 4). 

It is clear that, whatever the drug is, the increase of φ results in a reduction of the *D*/*D*_0_ ratio, while a reduction of the *r*_s_/ξ ratio causes an increase of *D*/*D*_0_ at fixed φ. It is important to underline that both φ and ξ variation, within the experimental field of this study, causes considerable variations of *D*/*D*_0_ (up to 40%) that, in turn, affect the drug release kinetics. The same holds for *r*_s_, whose accuracy in estimation is fundamental to achieve a reliable evaluation of *D*. Experimentally, the Stokes radius, *r*_s_, is commonly obtained by hydrodynamics techniques. Centrifugation/sedimentation, size exclusion chromatography (SEC), and electrophoresis are routinely used [33,34,35]. Dynamic light scattering (DLS) and NMR spectroscopy-based techniques are other analytical methods generally employed to measure diffusion coefficients and derive the corresponding hydrodynamic radius from the Stokes-Einstein equation [36,37,38]. Alternatively, theoretical and molecular models, alone [39,40] or in combination with experimental data (e.g., small-angle X-ray scattering (SAXS), Small-angle neutron scattering (SANS), high-resolution NMR, X-ray crystallography, cryogenic electron microscopy (cryo-EM)) [41], can be chosen to predict the hydrodynamic properties of proteins, macromolecules, drugs, or nanoparticles. The HYDRO suite of algorithms [42] deserves special mention, which allows the calculation of hydrodynamic properties simply from medium or high resolution models (e.g., a coordinate file) with substantial accuracy. We report here, for comparison, the values of *r*_s_ calculated for some of the substances considered in Figure 5 using the HYDROPRO module, starting from atomic coordinates: Theophylline, *r*_s_ = 0.37 nm, vitamin B_12_, *r*_s_ = 0.84 nm; myoglobin, *r*_s_ = 1.86 nm; BSA *r*_s_ = 3.5 nm [43,44]. Thus, this tool is particularly suited to the design and screening of molecules when experimental information is missing or scarce. 

When the polymeric network is characterized by meshes of different sizes, ξ_i_ (1 ≤ i ≤ *n*, relative abundance, *A*_i%_/100), for each mesh class, Equation (10) reads:
(11)$$D_{i}=D_{0}\left(1-\frac{{2r}_{s}}{\xi_{i}}\right)\exp(-Y\frac{\varphi }{1-\varphi })$$

In order to evaluate the effect of Equation (11) on the drug release kinetics, a possible strategy implies (a) solving the mass balance equation on a one-, two-, or three-dimensional grid (depending on the nature of the release process to be described) and (b) assuming that the drug diffusion coefficient on the generic grid node is given by Equation (11). The probability of finding the *D*_i_ value corresponding to the mesh size ξ_i_ is equal to *A*_i%_/100. A simpler way to proceed is to assume that the mass flux, *J*, is the sum of *n* contributes, each one weighting for *A*_i%_/100:
(12)$$J=-\overset{n}{\underset{i=1}{\sum }}\left(\frac{A_{i\%}}{100}D_{i}\right)\nabla C=-\nabla C\overset{n}{\underset{i=1}{\sum }}\left(D_{0}\left(\frac{A_{i\%}}{100}-\frac{A_{i\%}}{100}\frac{{2r}_{s}}{\xi_{i}}\right)\exp(-Y\frac{\varphi }{1-\varphi })\right)=-\nabla {C\;D}_{0}\left(1-\frac{{2r}_{s}}{\xi }\right)\exp(-Y\frac{\varphi }{1-\varphi })$$
(13)$$\frac{1}{\xi }=\overset{n}{\underset{i=1}{\sum }}\frac{A_{i\%}}{100}\frac{1}{\xi_{\iota }}$$

Obviously, this simpler strategy implies that the drug diffusion coefficient is the one referred to as the averaged mesh size as witnessed by Equation (13). Also, in this case, however, Equation (10) (Y = 1) implies no negligible effects on the release kinetics as depicted in Figure 6 in the case of BSA.

This simulation was built up by solving the mass balance assuming a spherical matrix (radius 0.5 mm), an infinite delivery environment, a uniform initial drug (BSA) distribution, no interactions between the polymeric chains and BSA, and the ξ–φ couples corresponding to hydrogels 30A, 30B, and 20C (see Figure 3 and Table 4). It is evident that upon the ξ decrease and φ increase, a more and more delayed release kinetics occurs.

In order to compare our theoretical approach with experimental data, we studied the release of a commonly used model drug [32], myoglobin, from our 20C hydrogel (20% PVP, 0.5% alginate, irradiation times of 44 min). Briefly, a myoglobin loaded 20C cylindrical hydrogel (radius ≈ 0.92 cm, height ≈ 0.18 cm, initial myoglobin concentration of 2 mg/cm^3^) was suspended by a web in a stirred release environment containing 10 cm^3^ of distilled water at 37 °C. At fixed times, the myoglobin concentration was spectrophotemetrically detected (see Section 4.5 for more details) so that the release profile shown in Figure 7 was obtained. The experimental release kinetics was fitted by a classical model relying on Fick’s law in the presence of a finite volume environment and assuming that drug release occurs only in the axial direction (radial diffusion was retained as negligible due to the small lateral surface) [32]:
(14)$$M_{t}^{+}=1-\overset{\infty }{\underset{i=1}{\sum }}\frac{2\alpha \left(1+\alpha \right)}{1+\alpha +\alpha^{2}+q_{i}^{2}}e^{\left(-q_{i}^{2}t\frac{D}{L^{2}}\right)}$$where M_t_^+^ is the ratio between the amount of drug released at time t and the amount released after an infinite time, L is the gel semi-thickness, D is the myoglobin diffusion coefficient in the hydrogel while q_i_ is the non-zero positive roots of:
(15)$$tan\left(q_{i}\right)=-\alpha q_{i}\;\alpha =\frac{V_{r}}{SL}$$where *S* is hydrogel cross section while *V*_r_ is the release environment volume. Equation (15) was solved according to the bisection method fixing the tolerance to 10^−5^. Equation (14) best fitting (see solid line in Figure 7), assuming 100 terms in the summation (further terms revealed to be unnecessary), was statistically satisfactory (F(1, 18, 0,95) < 318) and yielded to *D* = (2.3 ± 0.13) × 10^−9^ cm^2^/s. Remembering that the myoglobin diffusion coefficient in pure water at 37 °C is equal to 1.16 × 10^−7^ cm^2^/s [32], it turns out that the ratio of *D*/*D*_0_ for myoglobin is equal to 0.0195. Inserting this ratio in Equation (10), considering the myoglobin radius (1.91 nm) and the ξ–φ couple pertaining to hydrogel 20C (ξ = 18.6 nm, φ = 0.36), Y has to be equal to 6.57. As this value lies in between typical values found in the literature [29,31] (2.8 < Y < 30) for similar drugs, we can conclude that the approach used to characterize the hydrogel network (mesh size and polymer volume fraction) is reliable. 

## 3. Conclusions

The novel character of this work lies on the attempt of performing a global theoretical (mathematical) approach devoted to the evaluation of drug release from hydrogel based delivery systems in which the release kinetics is essentially ruled by drug diffusion inside the polymeric network. For this purpose, the interpenetrated polymeric network constituted by alginate and poly(*N*-vinyl-2-pyrrolidone) was considered because it is an interesting case study due to the lack of information about both the average mesh size (ξ) of the resulting network and the real polymer volume fraction (φ). Indeed, while ξ is usually unknown, φ is usually known. In this frame, the experimental outcomes descending from the combined use of rheology and LF-NMR proved to be a very useful strategy for the determination of the network characteristics. Indeed, the interpretation of these outcomes by means of proper theoretical models [24,25,26,27] allowed the evaluation of ξ and φ. In turn, ξ and φ knowledge allowed an estimation of the drug diffusion coefficient inside the polymeric network (*D*) to be formed according to the model proposed by Lusting and Peppas assuming Y = 1 [30]. Then, the embedding of *D* in the classical mass transport equation based on Fick’s law permitted the theoretical evaluation of the effect of ξ, φ, and *r*_s_ (solute radius) on drug release. Finally, in order to prove the reliability of the entire approach, experimental release data regarding myoglobin release from one of the studied hydrogels (20C) were fitted by means of Fick’s second law accounting for a finite release environment. Accordingly, the myoglobin diffusion coefficient was estimated for inside the hydrogel and, thus, the ratio of *D*/*D*_0_ (*D*_0_ is the myoglobin diffusion coefficient in water). It was verified that the Lustig-Peppas equation (Equation (10)) yields the same *D*/*D*_0_ ratio (0.0195) provided that Y = 6.57, thus it was concluded that the entire approach is reliable as, for similar drugs, Y ranges between 2.8 and 30 [29].

To our knowledge, a similar global modeling is not so common inside the pharmaceutical field of drug delivery.

## 4. Materials and Methods

Poly(*N*-vinyl-2-pyrrolidone) (PVP K90 ~3.6 × 10^5^ Da), hydrogen peroxide (30% wt) and sodium alginate (~10^6^ Da; high α-l-guluronic acid content ~70%) were provided by Sigma-Aldrich (Saint Louis, MO, USA). All the materials were used as received from the supplier and no further purifications were performed.

### 4.1. Interpenetrating Polymeric Network (IPN) Preparation

Polymer solutions were prepared by dissolving PVP and alginate in mixtures of Hepes 2 (10 mM HEPES and 150 mM NaCl in Milli-Q water, resulting in 98% v/v) to which H_2_O_2_ (2% v/v) was added under 250 rpm, stirring at 50 °C, until a homogeneous mixing was achieved. H_2_O_2_ plays the role of the photoinitiator. The UV radiation splits the H_2_O_2_ molecule into OH radicals which attack the polymer chains generating macroradicals. These latter propagate the reaction, creating covalent bonds (cross-linking points), which build up the polymer network. Solutions were prepared fixing the PVP concentration (% mass fraction) at 20% and 30% while the alginate concentration (% mass fraction) was always set at 0.5%. After preparation, polymer solutions were poured into syringes and left for 24 h at room temperature for the removal of bubbles.

IPN formation required crosslinking of PVP and, then, alginate. About 1 mL of polymeric blend solution was cast in cylindrical molds (~1 mm in depth, ~35 mm in diameter) on the top of which 0.5 mm thick glass discs were clamped to prevent water evaporation during crosslinking (we verified that glass absorbs a negligible amount of the radiation emitted by the UV lamps). The molds were placed at 15 cm from the UV source and irradiated for 22, 33, and 44 min (samples labeled A, B, and C, respectively) in a UV light chamber (BS-02, Dr. Gröbel UV Elektronik, Ettlingen, Germany) equipped with 4 tubes emitting in the UVB range (265–400nm) and 4 tubes in the UVC range (single peak at 253.4 nm). As the emitted power density was equal to 29 mW/cm^2^, samples A, B, and C received, respectively, an irradiated energy equal to 67 J, 100 J, and 134 J. Subsequently, hydrogel samples were removed from the molds with a spatula and placed in a Petri dish, where fixed volumes of a CaCl_2_ aqueous solution ([Ca^++^] = 9 g/L) were sprayed onto their surface to get alginate ionotropic gelation. Alginate crosslinking was allowed to last for 5 min. After crosslinking, hydrogels were dipped in deionized water under stirring to extract un-crosslinked polymer and photoinitiator (washing).

### 4.2. Rheology

The rheological characterization of hydrogels was carried out at 25 °C using a controlled-stress rheometer RS-150 (ThermoHaake, Karlsruhe, Germany) equipped with parallel plates (PP20Ti, diameter 20 mm) with serrated surfaces to avoid possible wall slippage, and provided with a Haake-F6 thermostat for temperature control. The measuring device was kept inside a glass bell at saturated humidity conditions to avoid evaporation effects. Hydrogels samples (~1 mm thick) were removed with the aid of a small spatula from the Petri dish in which they were prepared and subsequently put on a wood surface in order to cut a cylinder of a 20 mm diameter. Then, the cylinder was placed on the lower plate of the measuring device. The upper plate was then lowered to make contact with the hydrogel surface. Gap setting optimizations were performed according to the procedure described elsewhere [45]. The viscoelastic properties of hydrogels were analyzed under oscillatory shear conditions by applying different procedures. Stress sweep tests (SS) were carried out at constant frequency (1 Hz) in order to determine the extension of the linear viscoelastic region and the pattern of the nonlinear viscoelastic response. Frequency sweeps (FS) were performed within the linear viscoelastic regime at constant stress (5 Pa) in the frequency (*f*) range of 0.01 to 2 Hz.

### 4.3. LF-NMR

The LF-NMR analysis was performed at 25 °C by means of a Bruker Minispec mq20 (0.47 T, 20 MHz, Karlsruhe, Germany). The determination of the average water protons’ transverse (spin-spin) relaxation time inside the samples (*T*_2m_) was performed according to the CPMG (Carr–Purcell–Meiboom–Gill) [46] sequence {90°[−τ−180°−τ(echo)]*n*−*T*_R_} with a 8.36 μs wide 90° pulse, echo time τ = 250 μs, and *T*_R_ (sequences repetition rate) equal to 5 s. The criterion adopted to choose *n* consisted in ensuring that the final FID (free induction decay) intensity corresponded to approximately 1% of the initial FID intensity. In the light of this acquisition strategy, *n* spanned between 235 and 540. Finally, each FID decay, composed by *n* points, was repeated 36 times (number of scans). 

### 4.4. TEM

Hydrogel specimens were examined by TEM (Philips EM 208 100 kV, Eindhoven, The Netherlands). In order to remove the water without damaging the polymeric structure due to liquid loss, a fixative step in glutaraldehyde (GTA) [47,48] was adopted. Indeed, GTA, forming covalent bonds with polymeric chains, stabilizes and stiffens the hydrogel structure. Once dehydrated, hydrogels were embedded into epoxy resin that underwent polymerization up to the formation of a solid hard block. Subsequently, the epoxy-block was sliced into thin sections by an ultra-microtome, placed on a copper grid and stained with acetate uranyl; this last chemical produces the highest electron density and image contrast as well as imparting a fine grain to the image due to the high atomic weight of uranium (238).

### 4.5. Release Test

Myoglobin from horse skeletal muscle (17.6 kDa, Sigma Aldrich) was chosen for its wide use in the pharmaceutical field as a model drug [32]. In order to avoid possible photolysis reactions induced by UV irradiation (PVP crosslinking), hydrogel drug loading was obtained by soaking overnight hydrogel 20C in a myoglobin solution of a 2 mg/mL concentration. After the attainment of equilibrium, hydrogel 20C was extracted from the solution and washed with water to remove the residual myoglobin adsorbed or precipitated on the hydrogel surface. The amount of incorporated myoglobin was evaluated by the balance between the weight of the hydrogel before (0.29 g, radius ≈ 0.77 cm, height ≈ 0.15 cm) and after (0.49 g, radius ≈ 0.92 cm, height ≈ 0.18 cm) loading.

Myoglobin loaded hydrogel 20C was suspended by a thin web in a release environment (*V*_r_) containing 10 mL of distilled water at 37 °C. Stirring was ensured by a magnetic stirrer at 100 rpm. At established time intervals, 3 μL of dissolution medium were collected to measure myoglobin concentration by a UV spectrophotometer (408 nm, path-lengths of 20 mm). Release experiments were performed in duplicate.

## Figures and Tables

**Figure 1 gels-05-00022-f001:**
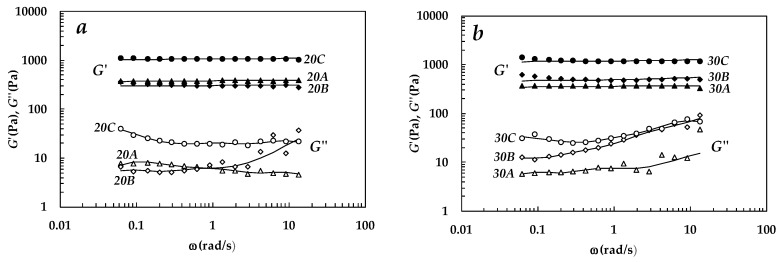
(**a**) Mechanical spectra referring to hydrogels, 20A (polymer mass percentage = 20.5% and 22 min irradiation), 20B (% polymer mass fraction = 20.5% and 33 min irradiation), and 20C (% polymer mass fraction = 20.5% and 44 min irradiation). (**b**) Mechanical spectra referring to hydrogels, 30A (% polymer mass percentage = 30.5% and 22 min irradiation), 30B (% polymer mass percentage = 30.5% and 33 min irradiation), and 30C (% polymer mass percentage = 30.5% and 44 min irradiation). Filled symbols represent the storage modulus, G′, while open symbols indicate the loss modulus, G″. Solid lines are the best fitting of the generalized Maxwell model (Equations (1)–(2)).

**Figure 2 gels-05-00022-f002:**
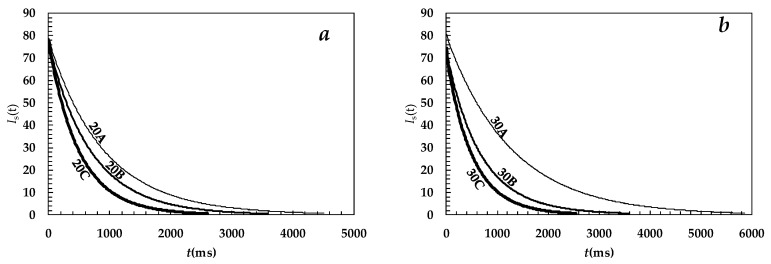
Experimental relaxation behavior of the modulus (*I*_s_) of the *xy* component of the magnetization vector referring to (**a**) hydrogels 20A (% polymer mass percentage = 20.5% and 22 min irradiation), 20B (% polymer mass percentage = 20.5% and 33 min irradiation), and 20C (% polymer mass percentage = 20.5% and 44 min irradiation); and (**b**) hydrogels 30A (% polymer mass percentage = 30.5% and 22 min irradiation), 30B (% polymer mass percentage = 30.5% and 33 min irradiation), and 30C (% polymer mass percentage = 30.5% and 44 min irradiation). Equation (5) best fitting is not visible as it practically coincides with the experimental data. *t* is time.

**Figure 3 gels-05-00022-f003:**
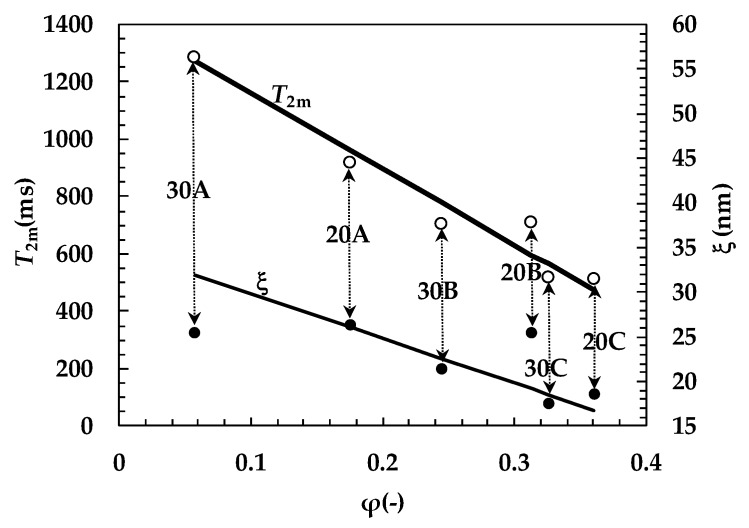
Dependence of the mean relaxation time of water hydrogens (*T*_2m_) and mesh size (ξ) on the polymer volume fraction, φ, referring to the different hydrogels considered (20A, 20B, 20C, 30A, 30B, and 30C). Solid lines indicate the linear interpolation of the experimental data.

**Figure 4 gels-05-00022-f004:**
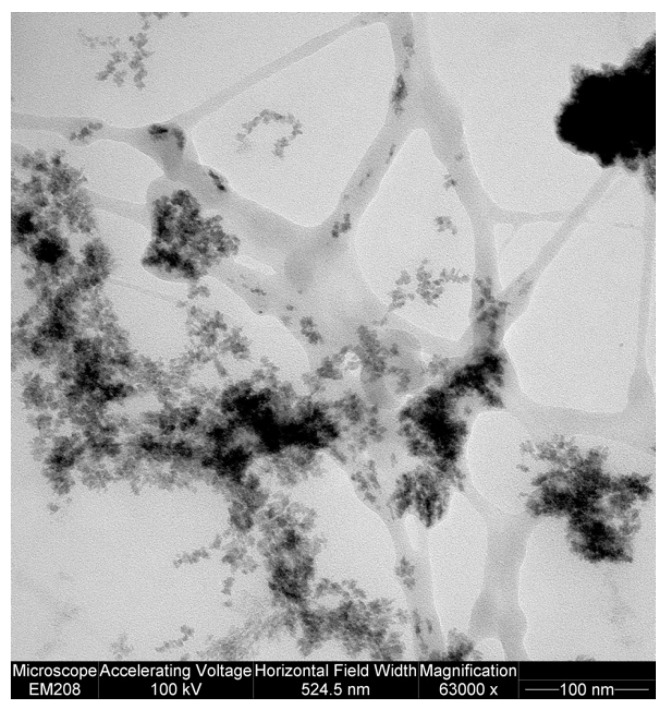
Transmission electron microscope (TEM) picture of hydrogel 20B. Gray ribbons represent polyvinylpyrrolidone (PVP) chains (the contrast agent weakly colors PVP chains), while dark zones represent crosslinked alginate strongly colored by the contrast agent.

**Figure 5 gels-05-00022-f005:**
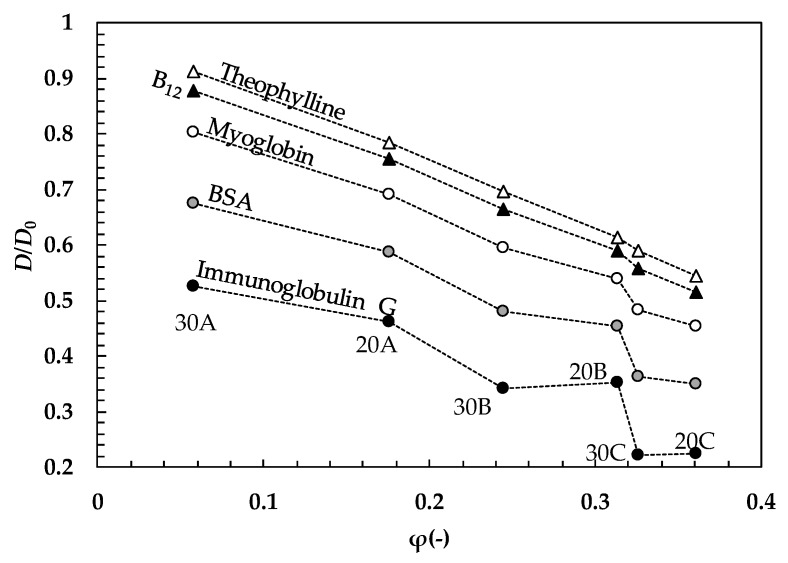
Variation of the drug diffusion coefficient (*D*) according to Equation (10) assuming Y = 1, drugs of different size (*r*_s_), and the couples of ξ–φ competing with the six studied hydrogels (Figure 3 and Table 4). *D*_0_ is the drug diffusion coefficient in pure water at 37 °C.

**Figure 6 gels-05-00022-f006:**
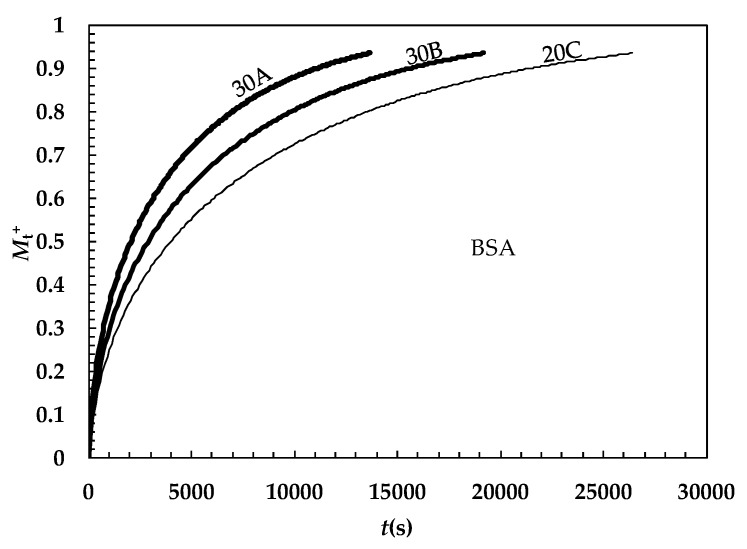
Time evolution of the dimensionless amount of bovine serum albumin (BSA) released from a spherical matrix (radius = 0.5 mm; *M*_t_^+^ = ratio between the amount of drug released at time t and the amount released after an infinite time). For BSA, *D*_0_ (37 °C, water) = 6.35 × 10^−8^ cm^2^/s [32].

**Figure 7 gels-05-00022-f007:**
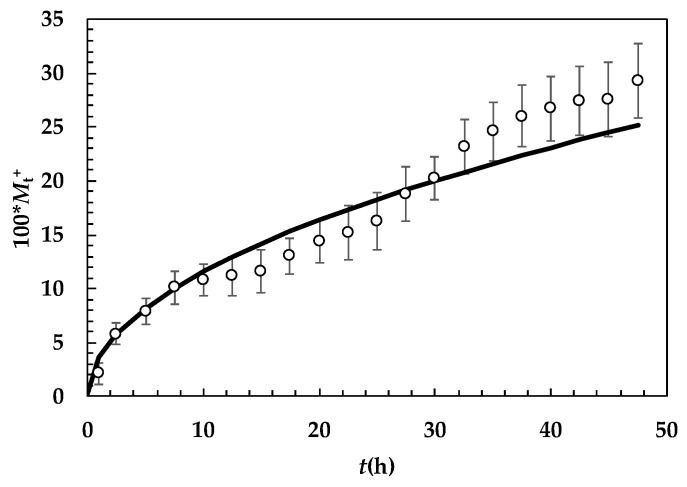
Time evolution of the dimensionless amount of myoglobin released from a cylindrical 20C hydrogel (*M*_t_^+^ is the ratio between the amount of drug released at time t and the amount released after an infinite time). Open circles indicate experimental data while the solid line is Equation (14) best fitting. Vertical bars represent datum standard error.

**Table 1 gels-05-00022-t001:** Generalized Maxwell model (Equations (1)−(2)) fitting parameters referring to the data (*G*′ and *G*″) shown in Figure 1a,b. While λ_1_ indicates the first Maxwell element relaxation time, *g*_i_ represents the spring constant value of the ith Maxwell element. Finally, *g*_e_ represents the purely elastic element.

System	λ_1_ (s)	*g*_1_ (Pa)	*g*_2_ (Pa)	*g*_3_ (Pa)	*g*_4_ (Pa)	g_e_ (Pa)
20A	(9.2 ± 2) × 10^−2^	8.2 ± 1.2	7.7 ± 1.4	15.1 ± 1.7	-	364 ± 18
20B	(5.9 ± 0.9) × 10^−2^	10.4 ± 0.8	11.1 ± 0.7	4.9 ± 0.9	17.3 ± 3.0	279 ± 9
20C	(10.0 ± 1) × 10^−2^	39.3 ± 1.3	25.5±1.5	19.5 ± 4.0	178 ± 23.0	854 ± 25
30A	(6.7 ± 3) × 10^−2^	13.0 ± 2	10.6 ± 2.0	9.1 ± 3.0	8.2 ± 8.0	337 ± 21
30B	(2.9 ± 2.6) × 10^−2^	200.6 ± 79	48.6 ± 26	16.0 ± 7.0	19.4 ± 7.6	453 ± 30
30C	(9.1 ± 1.1) × 10^−2^	136.6 ± 6	30.0 ± 5.0	33.0 ± 7.0	95.5 ± 25	1041 ± 32

**Table 2 gels-05-00022-t002:** Shear modulus (*G*), crosslink density (ρ_x_), and average mesh size (ξ) referring to the six hydrogel considered in this paper.

System	*G* (Pa)	ρ_x_ (mol/cm^3^)	ξ (nm)
20A	395 ± 28	(1.7 ± 0.10) × 10^−7^	26.0 ± 0.6
20B	434 ± 8	(1.9 ± 0.03) × 10^−7^	25.0 ± 0.1
20C	1117 ± 35	(4.9 ± 0.15) × 10^−7^	18.6 ± 0.2
30A	432 ± 23	(1.9 ± 0.10) × 10^−7^	25.5 ± 0.5
30B	737 ± 89	(3.2 ± 0.40) × 10^−7^	21.4 ± 0.9
30C	1335 ± 42	(5.8 ± 0.20) × 10^−7^	17.5 ± 0.2

**Table 3 gels-05-00022-t003:** Equation (5) fitting parameters (*A*_i%_, *T*_2i_), average relaxation time, *T*_2m_, and average inverse relaxation time (1/*T*_2_)_m_ (Equation (6)).

System	*A* _1%_	*T*_21_ (ms)	*A* _2%_	*T*_22_ (ms)	*T*_2m_ (ms)	(1/*T*_2_)_m_ (ms^−1^)
20A	36 ± 1.5	1162 ± 14	64 ± 1.5	782 ± 5	920 ± 22	(11.3 ± 0.2) × 10^−4^
20B	80 ± 1.1	771 ± 3	20 ± 1.1	449 ± 10	708 ± 11	(14.8 ± 0.3) × 10^−4^
20C	58 ± 1.1	595 ± 2	42 ± 1.1	393 ± 7	509 ± 8	(20.0 ± 0.4) × 10^−4^
30A	68 ± 17.0	1349 ± 36	32 ± 17	1141 ± 79	1283 ± 310	(7.8 ± 2.0) × 10^−4^
30B	26 ± 1.2	1029 ± 36	74 ± 1.2	587 ± 79	703 ± 16	(15.1 ± 0.3) × 10^−4^
30C	79 ± 2.0	571 ± 5	21 ± 2	321 ± 159	518 ± 15	(20.4 ± 0.9) × 10^−4^

**Table 4 gels-05-00022-t004:** Mesh size distribution (*A*_i%_, *T*_2i_) corresponding to the six studied hydrogels.

System	*A* _1%_	ξ_1_ (nm)	*A* _2%_	ξ_2_ (nm)	ξ (nm)	*R*_f_ (nm)
20A	36 ± 1.5	39.6	64 ± 1.5	22.1	26.0 ± 0.6	3.8
20B	80 ± 1.1	30.3	20 ± 1.1	15.4	25.0 ± 0.1	5.1
20C	58 ± 1.1	23.7	42 ± 1.1	14.4	18.6 ± 0.2	4.1
30A	68 ± 17.0	28.2	32 ± 17	21.2	25.5 ± 0.5	2.0
30B	26 ± 1.2	39.4	74 ± 1.2	18.4	21.4 ± 0.9	3.7
30C	79 ± 2.0	21.1	21 ± 2	10.8	17.5 ± 0.2	3.6

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
