# Peer review of "Theoretical Importance of PVP-Alginate Hydrogels Structure on Drug Release Kinetics"

_gels, 2019, doi:10.3390/gels5020022_

Round 1

Reviewer 1 Report

This manuscript describes how a combination of dynamic rheology and NMR can be used to analyze the hydrodynamic mesh size and polymer volume fraction within gels; specifically, those based on interpenetrating networks of UV-crosslinked PVP and calcium-crosslinked alginate. These findings are then used to make model predictions of the diffusivities of various biomolecules and rates of bovine serum albumin release. While using relatively simple rheology and NMR experiments to predict release rates is a worthy objective, the authors (regrettably) never experimentally verify that their model predictions are accurate. Further, the rheological and NMR methods used for structural characterization have already been reported by the authors’ group in Marizza et al. (Int. J. Polym. Mater., 2016) and Fanesi et al. (Pharm. Res., 2018), where the same techniques were applied to examine UV-crosslinked PVP-based gels (including those that used mixtures of PVP and alginate), which means the structural analyses reported herein do not appear to be novel either. Thus, though the general approach seems promising, this manuscript (in my opinion) provides little new value to the community without experimental D/D0 and release profile data that confirms that the authors’ microstructure-based predictions of release rates are indeed accurate. This work will be worth reviewing again once this experimental data is added.

Additionally, the way the Results and Discussion section is written, the reader must constantly interrupt their reading to flip to the Materials and Methods section. To make this story easier to read, experimental methods and model equations should be summarized in the Results and Discussion section sufficiently well for the work to be understood without having to flip to the Materials and Methods section. Some specific comments on the manuscript are:

1) Lines 110-112: This sentence should not be a separate paragraph. I recommend combining Figs. 1 and 2 (into side-by-side Figs. 1a and b) and combining the paragraphs that precede these plots.

2) Line 119: “being the storage (G’) modulus substantially” should be “being that the storage (G’) modulus is substantially”

3) Figs. 3 and 4: I recommend combing these figures into a single figure with two side-by-side plots.

4) Line 228: Here, the authors discuss the TEM images, without explaining how the gel samples were processed. To help the reader understand the result, it would be useful to briefly explain what was done to the samples prior to imaging. Further, in the Materials and Methods section at the end of the article, we learn that the gels were crosslinked further with glutaraldehyde prior to drying (to prevent structural changes upon dehydration). Does the glutaraldehyde crosslinking not affect the gel structure? It would be useful to show supporting information that shows that these potential changes are not problematic.

5) Lines 241-244: If the D/D0 values were predicted based on Equation 6, the authors should explicitly state that this was the case. If not, they should describe how D/D0 measurements were made.

6) Figure 7: It would be useful to compare the experimental data in this figure to the model predictions. These comparisons are typically made in similar studies.

7) Figure 8: In proving the model effective, the predicted release profiles should be compared with experimental ones. Leaving this experimental data out significantly weakens the story.

8) Conclusions section: The Conclusion section requires substantial revision. The authors state that their work “evidences the necessity of an overall theoretical approach for the delivery of drug release from hydrogel-based drug delivery systems.” However, I could identify no strong evidence for this “necessity” from the authors’ work. The Conclusion (which presently comprises only two sentences) should be sharpened to provide specific information that can be drawn from the authors’ work, while highlighting the novel aspects revealed in the present study.

9) Lines 304-305: Please indicate whether/how the materials were purified.

10) Line 308: What was the role of H2O2?

11) Equations 10-13: For clarity, these equations would have been useful to introduce in the Results and Discussion section.

Some other minor comments:

12) Line 46: Since chitosan, alginate and hyaluronan are all polysaccharides, the “and polysaccharides” is redundant.

13) Line 59: “poly-dispersed” should be “polydisperse”

14) Lines 121, 123, 253: “reflects” should be “results”

15) Line 156: “irrelevant” should be “insignificant”

16) Line 178: “relaxation time inverse” should be “inverse relaxation time”

17) Line 204: “inter correlations” should be “intercorrelations”

18) Line 210: “hydrogens relaxation” should be “hydrogen relaxation”

19) Line 212: “counter acting” should be “counteracting”

20) Line 254: “implies an increase” should be “causes an increase”

21) Line 255: “implies not negligible” should be “causes non-negligible”

22) Line 313: “implied to firstly” should be “required to first”

23) Line 322: “by the aid of” should be “with”

Line 328: “by means of” should be “using”

Author Response

Referee 1

0.1)the authors (regrettably) never experimentally verify that their model predictions are accurate.

In the light of this comment, the findings of our theoretical approach have been compared to experimental data regarding myglobin release from our 20C system. In particular, the myoglobin diffusion coefficient (D) inside system 20C was determined by the new eq.(14) fitting to the experimental release data shown in the new Figure 7. This allowed the evaluation of the D/D0 ratio, being D0 the myoglobin diffusion coefficient in water at 37°C. In turn, this ratio was used to evaluate the parameter Y appearing in the Lustig-Peppas model (eq.(10)), relaying on the knowledge of the mesh size (x) and the polymer volume fraction (f) referring to our system 20C (see Table 4 and Figure 3). As the resulting Y value (6.57) lies in the typical range found in literature (2.8 < Y < 30) for solute and polymers not too dissimilar from ours (Amsden, Macromolecules 1998, 31, 8382-8395; Dal Moro et al., Advances in Polymer Technology, 2012, 31, 219–230), we conclude that our theoretical approach is, at least, reasonable.

0.2) Further, the rheological and NMR methods used for structural characterization have already been reported by the authors’ group in Marizza et al. (Int. J. Polym. Mater., 2016) and Fanesi et al. (Pharm. Res., 2018), where the same techniques were applied to examine UV-crosslinked PVP-based gels (including those that used mixtures of PVP and alginate), which means the structural analyses reported herein do not appear to be novel either.

Referee 1 is absolutely correct as the approach shown in our paper has been adopted in our references 20 and 22. However, we do not believe that it is a negative aspect but a confirmation of the approach reliability. In addition, it is also true that in ref 20 we studied the same interpenetrated system but in this paper we enlarged our focus considering different PVP concentration and different irradiation time. Thus, this is another aspect supporting the reliability of our approach.

0.3) Thus, though the general approach seems promising, this manuscript (in my opinion) provides little new value to the community without experimental D/D0 and release profile data that confirms that the authors’ microstructure-based predictions of release rates are indeed accurate. This work will be worth reviewing again once this experimental data is added.

We believe to have properly answered to this issue by adding the release data shown in the new Figure 7 and the following interpretation relying of our approach and the Lustig-Peppas equation.

0.4) Additionally, the way the Results and Discussion section is written, the reader must constantly interrupt their reading to flip to the Materials and Methods section. To make this story easier to read, experimental methods and model equations should be summarized in the Results and Discussion section sufficiently well for the work to be understood without having to flip to the Materials and Methods section.

We strongly agree with this issue but this organisation was adopted to follow the template we downloaded from the joiurnal web site. Thus, we are more than happy to substantially modify paper organisation.

1) Lines 110-112: This sentence should not be a separate paragraph. I recommend combining Figs. 1 and 2 (into side-by-side Figs. 1a and b) and combining the paragraphs that precede these plots.

Figure 1 and Figure 2 have been merged together into the new Figure 1 a-b. In addition, the two paragraphs have been combined into only one preceding the two figures.

2) Line 119: “being the storage (G’) modulus substantially” should be “being that the storage (G’) modulus is substantially.

The replacement has been performed.

3) Figs. 3 and 4: I recommend combing these figures into a single figure with two side-by-side plots.

The merging has been performed.

4) Line 228: Here, the authors discuss the TEM images, without explaining how the gel samples were processed. To help the reader understand the result, it would be useful to briefly explain what was done to the samples prior to imaging. Further, in the Materials and Methods section at the end of the article, we learn that the gels were crosslinked further with glutaraldehyde prior to drying (to prevent structural changes upon dehydration). Does the glutaraldehyde crosslinking not affect the gel structure? It would be useful to show supporting information that shows that these potential changes are not problematic

As reported in the Materials and Method section, our hydrogels underwent a fixative step in glutaraldehyde in order to prevent gel damaging upon the subsequent dehydration. Once dehydrated, hydrogels were embedded into epoxy resin that underwent polymerization up to the formation of a solid hard block. Subsequently, the epoxy-block was sliced into thin sections by an ultra-microtome, placed on a copper grid and stained with acetate uranyl; this last chemical produces the highest electron density and image contrast as well as imparting a fine grain to the image due to the high atomic weight of uranium (238).

For what concerns the issue regarding the effect of the fixative step in glutaraldehyde, we recall that this is the standard method adopted to prevent dehydration damaging in living tissues (Desmon Key, Techniques for electron microscopy, Blackwell scientific publication Oxford and Edinburg 1965; Sabatini DD, Bensch K, Barnett RJ. Cytochemistry and electron microscopy. The preservation of cellular ultrastructure and enzymatic activity by aldehyde fixation. J. Cellular Biology 1963, 17, 19–58). Thus, we believe that the probability of getting artifacts/damages is much higher when the fixative step is not performed. In the light of this comment, the above mentioned references have been added.

5) Lines 241-244: If the D/D0 values were predicted based on Equation 6, the authors should explicitly state that this was the case. If not, they should describe how D/D0 measurements were made.

As stated in the original test, the D/D0 ratio has been evaluated according to the old eq.(6), now eq.(10), considering drugs characterized by different radii and assuming the xf couples referring to the studied hydrogels (20A, 20B, 20C, 30A, 30B, 30C):

“In the light of eq.(6), Figure 7 shows the variation of the ratio D/D0 assuming drugs of different dimension (theophylline, rs = 0.39 nm; vitamin B12, rs = 0.86; myoglobin, rs = 1.89; BSA rs = 3.6 nm; immunoglobulin G rs = 5.63 [30]) and the ξ - φ couples competing to our six hydrogels (see Figure 5 and Table 3)”. This information was also reported in the caption to the old Figure 7. In the light of this comment the text before the new Figure 5 and its caption was modified a little (see red text).

6) Figure 7: It would be useful to compare the experimental data in this figure to the model predictions. These comparisons are typically made in similar studies.

We agree with the referee about the necessity of comparing our model with experimental data. However, while this comparison has been added at the end of the paper (se answer to issue 0.1), the aim of the old Figure 7 was to underline the theoretical effect of x and f, competing to the different hydrogels (20A, 20B, 20C, 30A, 30B, 30C), on the D/D0 ratio referring to different drugs. This witnesses that the x-f couple of our hydrogels can significantly affect D/D0 in the case of small and big solutes. Thus, we decided to postpone the comparison between our approach and the experimental data at the end of the paper.

7) Figure 8: In proving the model effective, the predicted release profiles should be compared with experimental ones. Leaving this experimental data out significantly weakens the story.

As we agree with this issue, a release model relying on the classical Fick equation assuming a finite release environment volume has been fitted to the new experimental data. However, we believe that, following the philosophy adopted also in the old Figure 7, it is useful to maintain this figure in order to appreciate the not negligible effect of different x-f couples on the release profile.

8) Conclusions section: The Conclusion section requires substantial revision. The authors state that their work “evidences the necessity of an overall theoretical approach for the delivery of drug release from hydrogel-based drug delivery systems.” However, I could identify no strong evidence for this “necessity” from the authors’ work. The Conclusion (which presently comprises only two sentences) should be sharpened to provide specific information that can be drawn from the authors’ work, while highlighting the novel aspects revealed in the present study.

In the original version, Conclusions were very short following what stated in the GELS template: “This section is not mandatory, but can be added to the manuscript if the discussion is unusually long or complex”. However, in the light of this issue and due to the presence of experimental data, Conclusions were revised (see red text).

9) Lines 304-305: Please indicate whether/how the materials were purified.

All the materials were used as received from the supplier. No further purification were performed. This statement was added in the revised text.

10) Line 308: What was the role of H2O2?

H2O2 plays the role of the photoinitiator. The UV radiation splits the H2O2 molecule into OH·radicals which attack the polymer chains generating macroradicals. These latter propagate the reaction creating covalent bonds (cross-linking points) which build up the polymer network.

This explanation was added in the materials and methods section 4.1.

11) Equations 10-13: For clarity, these equations would have been useful to introduce in the Results and Discussion section.

Done.

12) Line 46: Since chitosan, alginate and hyaluronan are all polysaccharides, the “and polysaccharides” is redundant.

Done

13) Line 59: “poly-dispersed” should be “polydisperse”

Done

14) Lines 121, 123, 253: “reflects” should be “results”

Done

15) Line 156: “irrelevant” should be “insignificant”

Done

16) Line 178: “relaxation time inverse” should be “inverse relaxation time”

Done

17) Line 204: “inter correlations” should be “intercorrelations”

Done

18) Line 210: “hydrogens relaxation” should be “hydrogen relaxation”

Done

19) Line 212: “counter acting” should be “counteracting”

Done

20) Line 254: “implies an increase” should be “causes an increase”

Done

21) Line 255: “implies not negligible” should be “causes non-negligible”

Done

22) Line 313: “implied to firstly” should be “required to first”

Done

23) Line 322: “by the aid of” should be “with”

Done

24) Line 328: “by means of” should be “using”

Done

Reviewer 2 Report

In this work the use of rheology and Low Field NMR are applied for providing information about the polymeric network characteristics. However, I think that the study with only one type of gel (PVP and alginate in two different proportions and three different irradiation times) does not provide enough information to establish a theoretical approach of the role of hydrogels structure on delivery properties. In my opinion, the title and the summary are very ambitious for the results obtained.

Please, change the title and do mention PVP and alginate in the abstract.

Author Response

Referee 2

1) In this work the use of rheology and Low Field NMR are applied for providing information about the polymeric network characteristics. However, I think that the study with only one type of gel (PVP and alginate in two different proportions and three different irradiation times) does not provide enough information to establish a theoretical approach of the role of hydrogels structure on delivery properties. In my opinion, the title and the summary are very ambitious for the results obtained.

In the light of this comment, paper title has been changed and the abstract has been revised here and there (see red text).

Reviewer 3 Report

The manuscript by Michela Abrami et al. is a clear demonstration of the need of theoretical approach for properly describing the drug release from delivery systems based on hydrogel network. The work is well organized and presented and give adequate evidence of authors’ hypothesis. The experimental set up is adequate for the scope of the study and the conclusions are well supported by the obtained and discussed results. Thus, publication of the study in its present for is encouraged.

Author Response

Referee 3

1) The manuscript by Michela Abrami et al. is a clear demonstration of the need of theoretical approach for properly describing the drug release from delivery systems based on hydrogel network. The work is well organized and presented and give adequate evidence of authors’ hypothesis. The experimental set up is adequate for the scope of the study and the conclusions are well supported by the obtained and discussed results. Thus, publication of the study in its present for is encouraged.

We thank the referee for this comment.